# Safety of a Porous Hydroxyapatite Bone Substitute in Orthopedics and Traumatology: A Multi-Centric Clinical Study

**DOI:** 10.3390/jfmk9020071

**Published:** 2024-04-11

**Authors:** Leo Massari, Achille Saracco, Sebastiano Marchesini, Edoardo Gambuti, Alessandro Delorenzi, Gaetano Caruso

**Affiliations:** 1Department of Translational Medicine and for Romagna, University of Ferrara, c/o “S. Anna”, Via Aldo Moro 8, 44124 Ferrara, Italy; 2Department of Neurosciences and Rehabilitation, University of Ferrara, c/o “S. Anna”, Via Aldo Moro 8, 44124 Ferrara, Italy

**Keywords:** traumatology, orthopedics, hydroxyapatite, bone grafts, bone substitute, ORIF

## Abstract

The development of biomaterials in recent years has made it possible to broaden their use in the surgical field. Although iliac crest bone graft harvesting currently remains the gold standard as an autograft, the properties of hydroxyapatite bone substitutes appear to be beneficial. The first fundamental step to consider is the safety of using these devices. The purpose of this retrospective cohort study is to consider all the adverse events observed in our population and assess their relationships with the bone substitute device. The population analyzed consisted of patients undergoing trauma osteosynthesis with at least one implanted porous hydroxyapatite device. We considered a court of 114 patients treated at “Azienda Ospedaliera Universitaria di Ferrara—U.O. di Ortopedia e Traumatologia” in the period from January 2015 to December 2022. Upon analyzing our population, no adverse events related to the device emerged. Taking into consideration different study groups from other National Hospital Centers, no critical issues were detected except for three cases of extrusion of the biomaterial. It is necessary to clarify that bone substitutes cannot replace compliance with the correct principles linked to the biomechanics of osteosynthesis. This report outlines a safety profile for the use of these devices as bone substitutes in trauma orthopedic surgery.

## 1. Introduction

The development of surgical techniques has been accompanied by the improvement of the materials used in this field. Although the usefulness of bone grafts has long been known [1,2], during the last few years, interest in hydroxyapatite-based bone substitutes has grown considerably due to the considerable convenience of these devices [3]. Furthermore, there has been increasing attention to the management of bone defects, as they are closely linked to the fracture-healing process [4,5].

In the past, autologous grafts, mainly consisting of bone harvests from the iliac crest, represented the only solution for addressing fill defects or losses of bone substance [6,7].

These grafts, in fact, offer the best osteoinductive and osteoconductive properties, making them a reference for evaluations of effectiveness; however, they present some limitations consisting mainly of morbidity at the donor site (pain, iliac crest fractures, and hematoma) [8,9]. In the literature, acute pain has been reported for 2.8% to 27.9% of patients [10], and chronic pain has been reported for 2.4% to 60% of patients [11,12]. For this reason, numerous studies have been carried out to find valid substitutes for autologous grafts [13].

The properties of bioactive materials such as hydroxyapatite have been extensively analyzed to obtain devices capable of providing clinical benefits [14]. Their main use concerns conditions in which there is a need to fill areas where bone substance has been lost and improve the ossification healing process [15].

There are other bioactive materials on the market, such as hydroxyapatite, tricalcium phosphate, calcium phosphate, and calcium sulfate. These biomaterials have been widely studied for their ability to form direct bonds with bone, filling any defects in the substance. Obviously, they have different mechanical and biodegradability properties and characteristics.

Tricalcium phosphate and calcium sulfate are biomaterials with excellent structural properties and suitable osteoconductive properties, but they do not have osteoinductive and osteogenic capabilities. Among the above-mentioned biomaterials, calcium phosphate has the best structural qualities and is comparable to the previously mentioned ones with regard to osteoconductivity. It also has no osteoinductive or osteogenic capabilities. Furthermore, in clinical practice, it is also necessary to consider the different costs: there is a clear economic disadvantage for calcium phospate [16].

In the study presented, preformed bone substitutes based on hydroxyapatite (HA) are taken into consideration. HA is an important natural component of bone and can combine with tissues via chemical bonds to form a new bone matrix. This product is completely biomimetic because its chemical and structural characteristics make it completely like human bone [17]. The advantage of using HA compared to other biomaterials is that it has all the biological properties that can be used in a bone graft. Although it is not the material with the best absolute values in terms of individual characteristics, it represents an extremely useful compromise for surgical use. Table 1 delineates the characteristics previously taken into consideration.

In this retrospective study, a preformed product called ENGIpore (Fin-ceramica S.p.A, Faenza RA, Italy) was used. This product is ready to use and available in different formats (i.e., chips, wedges, blocks, hollow cylinders, and SH, a specific conformation for the proximal humerus). This product is an implantable, resorbable, non-active medical device acting as a bone substitute. It consists of biomimetic porous hydroxyapatite, which is very similar in its chemical composition and microstructure to the mineral component of human bones. These medical devices are characterized by a very high porosity, which can reach 90% relative to the total volume. Their interconnected pore systems are mainly composed of macro- and micro-pores. The size of these macro-pores ranges mainly between 200 and 500 μm, while the size of the interconnected micro-pores ranges between 80 and 200 μm. Moreover, these devices feature an intergranular microporosity (<10 micron), which facilitates the absorption of physiological fluids. This structure allows the device to optimize its colonization by cells predisposed to bone tissue regeneration, thus guaranteeing valid performance with regard to osteointegration and osteoconduction properties. A further feature is the long remodeling time of this device. These products begin to undergo changes in conformation approximately 24 months after their implantation. Thus, it is possible to guarantee long-lasting and reliable mechanical support during the healing period of a fracture.

There are numerous advantages to using these devices, such as the following:-They reduce the risk of disease transmission because they are easily sterilizable;-They do not induce a host inflammatory response;-They are versatile in use and can be supplied in different formats and conformations;-They are ready to use;-They have remarkable mechanical and biomimetic properties;-They eliminate the problems related to the donor site occurring with the use of autografts.

The instructions provided by the manufacturer for the use of these devices cover applications such as the following:-Treating bone defects of the axial and appendicular skeletal system;-Filling bone gaps and reconstructing damaged or excised areas;-Surgical treatments for pathological conditions such as primary or secondary degenerative diseases (e.g., pseudoarthrosis in non-critical defects), as well as corrective surgery for malformation diseases of the skeletal system;-Orthopedic, cranial, and spinal surgery applications, using techniques such as osteosynthesis and bone reconstruction, prosthetic or intervertebral implants, prosthetic revision, and spinal fusion.

The aim of this retrospective study was to take into consideration all the adverse events observed in our population and evaluate their possible relationships with the implanted hydroxyapatite device.

## 2. Materials and Methods

For this retrospective study, we considered a court of 114 patients (44 male; 70 female) treated surgically at “Azienda Ospedaliera Universitaria di Ferrara—U.O. di Ortopedia e Traumatologia” from January 2015 to December 2022.

Pediatric patients (with an age < 18) were excluded.

Three groups related to the fracture site were created: one for fractures of the proximal humerus, one for proximal tibia fractures, and one for calcaneus and astragalus fractures.

In this retrospective study, we considered the types of fractures, fracture sites, and typology of fixation devices.

All patients included in this study underwent ORIF (Open Reduction and Internal Fixation) surgery with the application of at least one pre-formed bone substitute EngiPore (Fin-ceramica S.p.A, Faenza RA, Italy). The devices used in our clinical practice were shaped like chips, wedges, blocks, and a specific shape for the proximal humerus called SH.

The data on the patients followed during the postoperative follow-up were retrieved, all of whom had undergone clinical and radiographic evaluation at 1 month and 3 months. The possible presence of clinical problems or radiographic signs was therefore assessed, and correlations with the device used were determined.

A clinical evaluation of the patients and their clinical symptoms was performed during the usual post-operative follow-up by members of our operating unit during the divisional outpatient visits. The retrospective evaluation of the radiographic images, however, was performed by two orthopedic surgeons with experience in the field of traumatology.

The data obtained were compared with those from other studies carried out at other Italian hospital centers by conducting assessments on numerically similar and therefore statistically comparable populations. The comparison populations had the same anamnestic characteristics and had been administered the same types of treatment. Thus, we were able to expand the population examined, obtaining more reliable and significant information.

### 2.1. Objective of the Study

The objective of this study was to research and report any adverse effects of the bioceramic matrix device. Our methods made it possible to obtain information regarding the safety of this device’s use in orthopedic traumatology. It is necessary to clarify that our evaluations were not intended to assess the effectiveness of bone substitutes.

### 2.2. Definition of an Adverse Event

All adverse events (AEs) encountered during follow-ups were reported. They have been described in a way that allowed us to report the following parameters [Appendix A]:-Description of an AE-AE status (closed or still ongoing)-Evolution/outcome (recovered, recovered with sequelae, death, or unknown)-Device explanted (NO or YES)-AE related to the use of the device (NO or YES)

All adverse events described in the analyzed papers were included in our analysis and categorized as shown in Appendix A.

## 3. Results

All 114 patients taken into consideration were treated surgically with the implantation of at least one bone substitute. The population was divided based on the skeletal segments involved, yielding the following outcomes as shown in Figure 1: -49 (43%) fractures of the tibial plateau and distal portion of the femur;-47 (41%) fractures of the proximal portion of the humerus;-18 (16%) fractures of the calcaneus or talus.

At our center, the ENGI-pore device, in its different conformations, has been used to manage multi-fragmentary fractures with the presence of bone gaps measuring at least approximately 0.5 cm^3^. For this reason, the entire device was not always used; sometimes, only portions of it were employed. In this way, the synthetic bone graft was shaped to the surgeon’s liking. In most situations, the device was used to treat fractures such as AO/OTA 11B, 11C, 33B, 33C, 41B2-B3, 41C, or 82C.

The average age of the patients was 55 ± 15.8 years. During the follow-up, 11 patients with different complications were found, as shown in Table 2.

We proceeded by analyzing these patients and evaluating any correlations between the adverse effects and the bone substitute used. No adverse effects could be attributed to the bone graft. Further attention was paid to patients with secondary breakdown of the fixation to evaluate any functional failures of the HA device.

In our cohort of patients, a frequency of generic adverse effects (AEs) comparable to the adverse effects observed in any population undergoing similar surgical procedures was observed. Furthermore, the AEs observed could not be clinically attributed to the presence of the device. For this reason, we interpreted this result as suggesting that the AEs were of a type inevitably present in this type of surgically treated population.

To obtain more clinical information, we chose to expand the study with a multicentric comparison. Therefore, we analyzed numerous data obtained from databases collected by other national centers, and we decided to proceed with a comparison between our results and those obtained in numerically similar populations hailing from the Emilia-Romagna region. All the patients considered had been treated for traumatic pathologies with ORIF and the same porous hydroxyapatite bone substitutes. Table 3 reports the samples included in the comparison, specifying origin, sample number, and adverse effects found.

## 4. Discussion

The growing use of bioceramic bone substitutes in traumatology is due to the numerous advantages that these products provide [18]. These devices offer considerable versatility of use as they are highly moldable, involve the use of a simple sterilization process, and have a valid safety profile.

The promising in vitro performance results reported make these devices valid alternatives in this field too. Currently, bone grafting from the iliac crest is the gold standard in terms of osteoinductive and osteoconductive properties [19]; however, it is not without adverse events, especially at the donor site.

In 2018, Turnbull G. et al. conducted a review with the aim of identifying the ideal properties that a 3D scaffold must present during its in vivo clinical use [20]. This review took into consideration various parameters both related to the production of the devices and their technical characteristics. Furthermore, the authors conducted an in-depth examination of the uses of the main products on the market. In this way, they provided valuable indications and models to investigate.

In the literature, several publications, including Lovati A.B. et al.’s [15], underline the validity of using bioceramic devices as bone substitutes. Furthermore, their ease of use and lower operating times and the absence of complications at the graft donor site prove to be in favor of their potential use in surgical practice. Also, Blom A W et al. [21] report encouraging data in the orthopedic field regarding the use of bioceramic materials in acetabular revisions. The cited authors performed a single-institution, multi-surgeon, prospective cohort study with 43 patients, of which 9 had cemented acetabular components implanted, and 34 had them uncemented. The use of bone chips allowed the management of bone defects. The use of bone chips made it possible to manage the loss of bone tissue by applying these bioceramic grafts specifically designed for use in impaction grafting. This study reported no side effects and also demonstrated good efficacy in short-term results.

In agreement with the studies by the authors mentioned above, there is the work of Cimatti P. et al. [22], which takes into consideration the use of these devices in hip revision surgery. The authors of this study performed a 2-year follow-up on 29 patients undergoing revision hip prosthesis surgery, using a collagen–hydroxyapatite composite biomaterial device for the management of bone loss at the acetabular level. Also, in this case, no serious postoperative complications emerged, and the results observed, both from a clinical and radiographic point of view, were considered satisfactory. Furthermore, good scaffold integration was observed.

However, for their proper use, it is fundamental to assess the safety of grafts by looking for post-operative complications.

Although it is essential to focus on major adverse effects as they are potentially responsible for the failure of surgical treatment, it is still important to monitor even minor forms of adverse effects to better understand the use of these devices.

In the orthopedic field, as well as in spinal surgery, there is more information regarding the use of biomaterial bone substitutes compared to that regarding traumatology. For this reason, some of these studies were taken into consideration to broaden our evaluation.

Barbanti Brodano G. et al. [23] conducted a post-market analysis on spinal surgery with a minimum follow-up of 12 months and a maximum of 5 years. No intra-operative or post-operative complications were observed in this study, thus outlining the valid safety profile of the hydroxyapatite devices used.

Recently, in 2022, Liu X.J. et al. [24] conducted a meta-analysis on spinal surgery to evaluate the effectiveness of hydroxyapatite devices. At the end of the selection process, 12 studies were deemed suitable. The result of this meta-analysis provoked the authors to state that the clinical application of HA and related composite materials in spinal reconstruction is comparable to that of autologous bone, with satisfactory efficacy and safety.

In the traumatology field, there are no comprehensive studies that take into consideration the safety and effectiveness of hydroxyapatite devices. However, the clinical studies conducted, such as the study conducted by Marcacci M. et al. [25], presented satisfactory results for long term follow-ups. Since the relevant publications have mainly focused on efficacy studies, we considered it appropriate to conduct this study starting from the fundamental assumption made for every medical device, namely, the safety of its use.

During the evaluation of the complications encountered in patients in the post-operative period, it was fundamental to determine whether these complications were related to the use of the bone substitute.

Some complications encountered concerned secondary fracture breakdown or the malalignment of the synthesis. In two patients with ORIF failure, a suboptimal initial synthesis was observed. In the comparison groups hailing from different national centers, three cases of extrusion emerged that can be considered correlated to the presence of the device.

Given the absence of AEs attributable to the HA device in our group and the few AEs observed in the comparison groups from other centers, it was not considered appropriate to perform statistical analyses.

It is essential to reiterate that the positioning of the bone substitute cannot be considered a substitute for respecting the correct principles of the biomechanics of osteosynthesis.

### Limitations and Advantages of the Study

This study has several limitations, mainly due to the fact that it is a retrospective cohort study. Furthermore, different patterns and different sites of fracture have also been taken into consideration. However, this is a multicentric clinical trial with a large population examined treated entirely with the same type of bone substitute.

## 5. Conclusions

Considering the data obtained from our population and the remaining study groups included, a safety profile of the bioceramic bone substitutes used was developed.

In detail, as previously reported, no adverse events were observed in the population coming from our reference center. Furthermore, the low percentages (1.23% and 3.92%, respectively) of major or minor complications observed at the two other centers examined outline a clear safety profile. This safety profile is also supported by the data reported in the scientific literature in similar application areas [26].

These conclusions allow any further studies regarding effectiveness to be structured such that the use of the devices is optimized and better reveal the interactions between the biomaterial and the surgical intervention.

### Future Directions

The safety profile of these hydroxyapatite devices, highlighted by this study, should be the starting point for conducting further evaluations regarding traumatology to confirm these data and, above all, obtain more information regarding effectiveness. Furthermore, with increasing frequency, the application of bone substitutes is becoming a typical surgical approach to the management of certain types of fractures, and there are already devices on the market with conformations dedicated to a particular anatomical district. Improving data collection and feedback from operators will allow the further refinement of these products.

## Figures and Tables

**Figure 1 jfmk-09-00071-f001:**
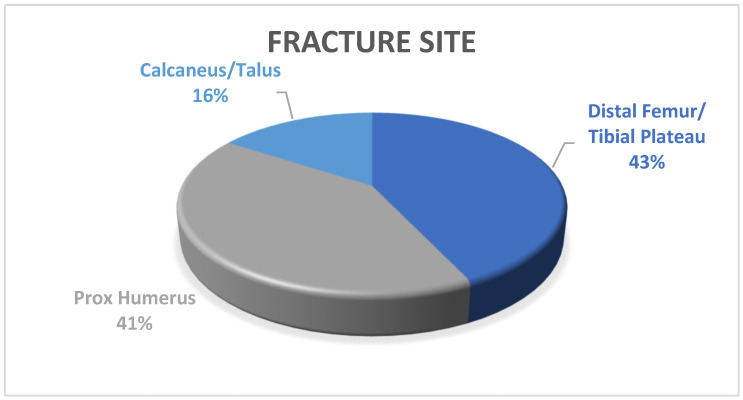
Distribution graph of fractures included in the population under consideration.

**Table 1 jfmk-09-00071-t001:** Properties of various forms of bone grafts.

Biomaterial	Osteoconductive	Osteoinductive	Osteogenic	Structural
Tricalcium Phosphate	+	-	-	++
Calcium Sulfate	+	-	-	++
Calcium Phosphate	+	-	-	+++
Hydroxyapatite	++	+	+	+

Key: + indicates good properties; ++ indicates excellent properties; +++ indicates the best properties; - indicates the absence of a given property.

**Table 2 jfmk-09-00071-t002:** Summary of the complications encountered.

Adverse Effect	Number of Patients (Percentage of the Total)	Bone Substitute Related (NO/YES)
Intolerance of fixation device	2 (1.75%)	NO
Wound-healing problems	2 (1.75%)	NO
ORIF failure	2 (1.75%)	NO
Persistent pain	2 (1.75%)	NO
Limited range of motion	2 (1.75%)	NO
Malalignment	1 (0.88%)	NO

**Table 3 jfmk-09-00071-t003:** Origin, number, and related AEs in the comparison groups.

Center	Number of Patients	Related AE (Percentage of the Total)
“M. Bufalini”—Cesena	121	0
“Maggiore”—Bologna	81	1 * (1.23%)
“Osped. Degli Infermi”—Rimini	65	0
“Policlinico”—Modena	51	2 * (3.92%)

* Device extrusion.

## Data Availability

The data presented in this study are available on request from the corresponding author.

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
