# Peer review of "Safety of a Porous Hydroxyapatite Bone Substitute in Orthopedics and Traumatology: A Multi-Centric Clinical Study"

_jfmk, 2024, doi:10.3390/jfmk9020071_

Round 1

Reviewer 1 Report

Comments and Suggestions for Authors

The paper present a retrospective study on the use of HA bioceramic bone substitutes in humans, focusing on the description of the side effects.

The paper needs to be improved for being considered for publication. 

The main point is you have examined HA bone substitute materials, but you didn't describe which products were actually tested. Could you give information about the composition of the implanted materials?

What about including an analysis of the of the initial dimensions of the fractures? 

The introduction is very poor. It should be widely expanded giving more details about the epidemiology of the need of bone fillings. Some parts lack specific information, such as line 52...what are the mechanical properties etc of the different materials. Give more information to the readers, some examples. Line 65 and following, please specify the fact that this is a retrospective study.

Line 216, even though statistical analyses were not done, you could describe the frequency of each observed side effect.

What about including some radiographs of a well healed fracture vs fractures that didn't?

How many assessors evaluated the radiographs?

Comments on the Quality of English Language

I didn't detect any major English mistake.

Author Response

Dear reviewer,

The paper present a retrospective study on the use of HA bioceramic bone substitutes in humans, focusing on the description of the side effects.

The paper needs to be improved for being considered for publication. 

Point 1: the main point is you have examined HA bone substitute materials, but you didn't describe which products were actually tested. Could you give information about the composition of the implanted materials?

Response: we have added more information about the bioceramics used to create these devices, also providing information in the product data sheet.

Point 2: What about including an analysis of the of the initial dimensions of the fractures? 

Response: the types of fracture treated and the minimum bone gap considered adequate for the implantation of the device were provided to the reader

Point 3: The introduction is very poor. It should be widely expanded giving more details about the epidemiology of the need of bone fillings. Some parts lack specific information, such as line 52...what are the mechanical properties etc of the different materials. Give more information to the readers, some examples. Line 65 and following, please specify the fact that this is a retrospective study.

Response: the introduction has been implemented, adding more information both on the device used and, on the properties, and general characteristics of bone substitutes. This provides a useful overview for making a valid comparison.

Point 4: Line 216, even though statistical analyses were not done, you could describe the frequency of each observed side effect.

Response: inserted in the tables the frequencies with which AE were found. The text contains a description of these AEs and a comparison with a population undergoing similar interventions without bone replacement grafting.

Point 5: What about including some radiographs of a well healed fracture vs fractures that didn't?

Response: I have not included radiographic images for the comparison between correctly healed fractures vs situations with adverse effects as this is exclusively a safety study. Therefore, I did not consider it appropriate to create misunderstandings regarding the effectiveness of the devices.

Point 6: How many assessors evaluated the radiographs?

Response: 2 Orthopedic Surgeons experienced in Traumatology.

In the hope that you will appreciate the paper, I send you my best regards.

Reviewer 2 Report

Comments and Suggestions for Authors

Surgical treatment of bone defects of various nature occupies a prominent place in clinical practice. In this regard, assessing the safety of bone substitutes based on hydroxyapatite, which is an analogue of hard tissue in mineral composition, is of significant interest. The authors of the article generally analyzed cases using porous hydroxyapatite materials. However, this assessment requires detailed confirmation with illustrative materials.

In total, the work contains 1 figure and 2 tables, in which all parameters for assessing safety of bone substitutes (for example, defect size, defect complexity, defect location) are not fully disclosed. The authors should pay attention to clinical cases where a negative result was observed. In addition, it is necessary to more clearly describe what size defects these implants were used for.

The section Introduction omits reference [13]. The section Discussion focuses heavily on general issues rather than on analysis of the clinical use of the materials.

In the section Materials and Methods, it is necessary to describe the characteristics of the bone substitute, the safety of which was assessed in this work. In addition, it is not clear whether there are any advantages to using this substitute compared to its analogues.

Author Response

Dear reviewer,

Surgical treatment of bone defects of various nature occupies a prominent place in clinical practice. In this regard, assessing the safety of bone substitutes based on hydroxyapatite, which is an analogue of hard tissue in mineral composition, is of significant interest. The authors of the article generally analyzed cases using porous hydroxyapatite materials. However, this assessment requires detailed confirmation with illustrative materials.

Point 1: In total, the work contains 1 figure and 2 tables, in which all parameters for assessing safety of bone substitutes (for example, defect size, defect complexity, defect location) are not fully disclosed. The authors should pay attention to clinical cases where a negative result was observed. In addition, it is necessary to more clearly describe what size defects these implants were used for.

Response: an additional table has been added to allow comparison with other biomaterials used as bone substitutes. The text also reports, through the AO/OTA classification, the types of fractures treated, the location and the minimum size of the bone defect treated with such devices. The clinical cases in which AE were found were followed up with more frequent follow up but, in our observation, no case was attributable to the presence of the bone substitute.

Point 2: The section Introduction omits reference [13]. The section Discussion focuses heavily on general issues rather than on analysis of the clinical use of the materials.

Response: The reference [13] has been added. In the introductory part and later in the discussion we increased the evaluation of clinical uses. However, it is necessary to specify that this is a safety-only study and that it is not our intention to provide advice on effectiveness.

Point 3: In the section Materials and Methods, it is necessary to describe the characteristics of the bone substitute, the safety of which was assessed in this work. In addition, it is not clear whether there are any advantages to using this substitute compared to its analogues.

Response: we have increased the description of the technical characteristics of the device and inserted a comparison table with the other biomaterials. In this way, for the reader, it is possible to have information about the advantages and disadvantages of HA compared to other products.

In the hope that you will appreciate the paper, I send you my best regards.

Round 2

Reviewer 1 Report

Comments and Suggestions for Authors

The authors improved their manuscript. Even though the study is quite simple, it does help increasing the available literature data on the safety of HA biomaterials in humans.

Comments on the Quality of English Language

No issues detected 

Author Response

Comments: The authors improved their manuscript. Even though the study is quite simple, it does help increasing the available literature data on the safety of HA biomaterials in humans.

Answer: We improved some passages of the text and corrected some sentences in English.

I hope you appreciate these improvements!

Reviewer 2 Report

Comments and Suggestions for Authors

In the Table 1 it is necessary to decipher the designations "-", "+", "++", "+++".

The corrected article may be published after minor revision and editorial revision of the quality of English.

Author Response

Comments: In the Table 1 it is necessary to decipher the designations "-", "+", "++", "+++".The corrected article may be published after minor revision and editorial revision of the quality of English.

Answer: Thank you for reporting this inaccuracy which has been corrected. We have also improved the English form.

I hope you appreciate these improvements